# Unraveling the Immunopathogenesis of Multiple Sclerosis: The Dynamic Dance of Plasmablasts and Pathogenic T Cells

**Yasunari Matsuzaka** [1,*] **and Ryu Yashiro** [2]

1. Division of Molecular and Medical Genetics, Center for Gene and Cell Therapy, The Institute of Medical Science, The University of Tokyo, Tokyo 108-8639, Japan
2. Department of Mycobacteriology, Leprosy Research Center, National Institute of Infectious Diseases, Tokyo 162-8640, Japan; ryuy@niid.go.jp
* Correspondence: yasunari80808@ims.u-tokyo.ac.jp; Tel.: +81-3-5449-5372

**Abstract:** Multiple sclerosis (MS) is a chronic inflammatory demyelinating disease of the central nervous system, characterized by multiple lesions occurring temporally and spatially. Additionally, MS is a disease that predominates in the white population. In recent years, there has been a rapid increase in the number of patients, and it often occurs in young people, with an average age of onset of around 30 years old, but it can also occur in children and the elderly. It is more common in women than men, with a male-to-female ratio of approximately 1:3. As the immunopathogenesis of MS, a group of B cells called plasmablasts controls encephalomyelitis via IL-10 production. These IL-10-producing B cells, called regulatory B cells, suppress inflammatory responses in experimental mouse models of autoimmune diseases including MS. Since it has been clarified that these regulatory B cells are plasmablasts, it is expected that the artificial control of plasmablast differentiation will lead to the development of new treatments for MS. Among CD8-positive T cells in the peripheral blood, the proportion of PD-1-positive cells is decreased in MS patients compared with healthy controls. The dysfunction of inhibitory receptors expressed on T cells is known to be the core of MS immunopathology and may be the cause of chronic persistent inflammation. The PD-1+ CD8+ T cells may also serve as indicators that reflect the condition of each patient in other immunological neurological diseases such as MS. Th17 cells also regulate the development of various autoimmune diseases, including MS. Thus, the restoration of weakened immune regulatory functions may be a true disease-modifying treatment. So far, steroids and immunosuppressants have been the mainstream for autoimmune diseases, but the problem is that this kills not only pathogenic T cells, but also lymphocytes, which are necessary for the body. From this understanding of the immune regulation of MS, we can expect the development of therapeutic strategies that target only pathogenic immune cells.

**Keywords:** anti-aquaporin 4; Blimp1; interleukin-10; experimental autoimmune encephalomyelitis; jun B proto-oncogene; multiple sclerosis; T helper 17 cell; programmed cell death-1; plasmablasts; receptor activator of NF-κB ligand

## 1. Introduction

Multiple sclerosis (MS) is a chronic inflammatory demyelinating disease of the central nervous system, characterized by multiple lesions occurring temporally and spatially [1]. However, the details of the mechanism of onset have not been clarified. The diagnosis is usually confirmed by demonstrating temporal and spatial multifocal lesions through detailed medical history and neurological examination over time and ruling out other diseases [2]. On the other hand, patients presenting with symptoms primarily of optic nerve and spinal cord origin include those with neuromyelitis optica spectrum disorders (NMOSD), originally known as a non-recurring disease with an inflammatory background that severely damages the optic nerve and spinal cord in a relatively short period of time [3]. However, in recent years, it has become clear that recurrent pathology is common, and

the involvement of serum anti-aquaporin 4 (AQP4) antibodies in pathogenesis is being elucidated [4]. Furthermore, among AQP4 antibody-positive patients, there are various patterns of NMOSD, such as those with lesions not only in the optic nerve and spinal cord but also in the brain; those with lesions only in the spinal cord or optic nerve; and those who are negative for AQP4 antibodies but have symptoms characteristic of NMOSD.

The currently used disease-modifying drugs can effectively reduce relapses but are not effective in primary progressive disease. Furthermore, it is impossible to halt the progression of the disorder once it enters the secondary progressive stage, even in the relapsing tolerant type. Therefore, it is considered important to diagnose and start treatment as early as possible. In this review, we summarize the genetic and environmental factors, pathological animal models, cellular and molecular mechanisms, and clinical treatment strategies of MS with our current understanding of the novel molecular mechanisms of pathogenesis (Table 1).

**Table 1.** Treatment strategy in MS.

| Treatment | |
| --- | --- |
| Steroid pulse therapy | methylprednisolone |
| Plasmapheresis therapy | |
| Post-treatment | oral steroids administered after steroid pulse therapy |
| Injection drug | IFNbeta1b, IFNbeta1a, glatiramer acetate, ofatumumab |
| Intravenous therapy | natalizumab |
| Oral medicine | fingolimod, dimethyl fumarate, siponimod |

The basic pathology of MS is "inflammatory demyelination" and many attempts have been made to find the cause of this inflammation [1,2]. On the other hand, with the advocacy of experimental autoimmune encephalomyelitis (EAE) and the subsequent induction of EAE by autoreactive lymphocyte transplantation [5], the hypothesis that "inflammatory demyelination" due to an autoimmune mechanism causes the onset of MS has been supported. Nevertheless, the disease-modifying effects of general immunosuppressive drugs are unclear, and interferon β (IFNβ) preparations were developed with the expectation of antiviral effects [6]. This was initiated by the idea that viral invasion of the central nervous system could be the etiology of MS. The administration of native IFNβ to patients with MS reportedly reduced relapses, subsequently suggesting that viral infections may be significantly associated with MS.

Although the pathogenesis of MS has not been fully elucidated, the following mechanisms are presumed. CD4+ T helper type 1 (Th1) cells activated by peripheral antigen presentation pass through the blood–brain barrier (BBB) and invade the central nervous system [7]. Th1 cells in central nervous system tissue are represented by antigen-presenting cells and reactivated to produce Th1 cytokines. B cells are differentiated by cytokines of Th1 and mature B cells produce antibodies against self-antigens, triggering an antigen–antibody reaction in the myelin sheath. Cytotoxic macrophages are also activated by Th1 cytokines, produce large amounts of inflammatory cytokines, and cause demyelination along with antibodies. Since the pathogenesis of MS is not clear, the mechanism of action of IFNβ against it has not been fully elucidated. However, it is thought to be effective against MS by (i) suppressing antigen presentation, (ii) inhibiting Th1 cell infiltration into the BBB, (iii) suppressing Th1 cytokine production, and (iv) exhibiting antiviral effects.

Biological data also suggest that the development of neutralizing antibodies in general is associated with reducing drug efficacy. In patients with MS, neutralizing antibodies of IFNβ are associated with markedly diminished biologic effects of IFNβ and reduced clinical efficacy. Neutralizing antibodies of IFNβ preparations in patients with MS often appear 6 months to 2 years after the start of administration, and their effects on clinical efficacy become noticeable even later. However, neutralizing antibodies often disappear during long-term treatment, so the discontinuation of administration requires careful consideration.

The production of neutralizing antibodies is affected by the drug structure, manufacturing and storage method, route of administration, dosage, and frequency of administration.

In addition, the most common adverse events associated with INFβ-1a administration were influenza-like symptoms such as fever, chills, headache, myalgia, asthenia, fatigue, nausea, and vomiting. A depressive state and decreased blood cell count were rarely observed. Patients with MS are known to have a higher incidence of depressive states such as depression than healthy adults, and depression and suicide attempts have been reported in patients receiving IFNβ preparations.

In clinical trials for patients with MS, INFβ-1a decreased the number of Gd-enhancing lesions and the volume of Gd-enhancing lesions detected by brain MRI examination, and suppressed the increase in new T2 lesions, enlarged T2 lesions, and the spread of MS lesions. It also reduced the recurrence rate and the number of annual intravenous steroid treatments and prolonged the time to the onset of the sustained progression of physical dysfunction.

Although the cause of MS is "unknown" by definition, many disease-modifying drugs have been developed as parts of various pathologies have been elucidated. As a result, the long-term prognosis has clearly improved, but therapeutic strategies for progressive MS are still insufficient. It goes without saying that the central nervous system is susceptible to irreversible sequelae, but the principle of MS treatment is early diagnosis and early treatment. On the other hand, there is also the risk of serious side effects occurring. In other words, while looking at the presence or absence of poor prognostic factors, the passive risk of progression of MD due to inadequate treatment is estimated, and the risk of side effects that may arise from the selection of disease-modifying drugs is considered. Personalized medicine is necessary for MS, which needs to optimize the risk–benefit balance for each patient.

## 2. Pathogenesis of MS

Records of retrospectively probable MS cases date back to the 14th century. The pathological concept was proposed by the founder of neurology, Jean Martin Charcot, in 1868, about 150 years ago [8]. Later, in 1965, the clinical definition of MS, namely "central nervous system demyelinating diseases of unknown cause showing spatial and temporal multifocality", was established, and the current clinical picture of MS was largely established. Although the cause of MS is not yet clear, lesions are infiltrated by lymphocytes and macrophages, and inflammation mediated by an autoimmune mechanism is thought to cause demyelination [9]. In the human body, the immune system, centered on white blood cells and lymphocytes, works to protect itself from external enemies such as bacteria and viruses. MS tends to be more common in areas and homes with good hygiene. One of the causes of MS may be an autoimmune attack on myelin triggered by infection with a herpes virus, when exposed to infectious agents such as viruses at an older age. Based on the hypothesis that viral infection is the etiology of MS, interferon b1b (IFNb1b), which has an antiviral effect, was tested for MS and approved in the United States as the first disease-modifying drug in 1993 [10].

On the other hand, in a group of patients called "opticospinal MS (OSMS)", treatment with IFNb1b often worsens their condition. In 2004, it was reported that a common autoantibody was detected in the sera of OSMS patients with optic neuritis and myelitis and Devick's disease and was called neuromyelitis optica (NMO)-Immunogloblin G (IgG) [11]. The antigen of this NMO-IgG was astrocyte AQP4. The clinical features and therapeutic responsiveness of disease groups in which anti-AQP4 antibodies are detected are different from those of MS. As a result, NMO-spectrum disorders (NMOSD) that can be diagnosed by antibody positivity are now clearly distinguished from MS.

## 3. Model Animal of MS

In addition, MS is more likely to develop in areas with short hours of sunlight, and low vitamin D levels during pregnancy and childhood due to lack of sunlight are factors that promote the development of MS [12]. Furthermore, MS belongs to the so-called

complex genetic diseases, has a moderate genetic risk, and is a disease with large number of cases affected by gene–environment interactions in various aspects. Thus, MS is a disease concept with ambiguous boundaries, and it is assumed that the cause is unknown. Therefore, it is theoretically impossible to establish a model animal that can serve as a golden standard for drug development. In practice, however, EAE models have often been used in drug development for MS [13]. The origin of EAE dates to 1933, when rabbit brain extracts were administered to monkeys in search of an animal model of acute disseminated encephalomyelitis (ADEM) after live vaccination, in which demyelinating lesions with spatial multifocality were detected [14]. "Rabbit brain extract" was replaced with myelin basic protein (MBP) and myelin-oligodendrocyte glycoprotein (MOG), a myelin sheath protein, as the immunogen, and the host animal was changed from monkeys to rodents, leading to the current EAE model, in which artificial autoimmunity is provoked by inoculation with immunostimulants and myelin antigens [15–17]. Although there is persistent criticism that EAE with a clearly established cause should not be used as a model for MS with an unknown cause, there are drugs that have been successfully developed for their efficacy against EAE. For example, glatiramer acetate is a random polypeptide composed of glutamic acid, lysine, alanine, and tyrosine that was created to mimic MBP, an immunogenic source of EAE, confirming the effect of suppressing the onset of EAE [18]. Additionally, the suppression of EAE by natalizumab, an anti-a4 integrin antibody, led to the blockage of lymphocytes from crossing the blood–brain barrier [19]. On the other hand, anti-TNF antibody strongly suppresses the onset of EAE, while the administration of lenercept, which inhibits the tumor necrosis factor (TNF) signaling system, induces relapse in MS patients. Thus, EAE seems to reproduce some of the pathology of MS, but due to the heterogeneity of MS itself, it is reasonable to consider that the pathological homology between MS and EAE is limited.

## 4. Symptoms in MS

The symptoms of MS vary greatly between individuals, depending on which nerve fibers are demyelinated (Figure 1) [20–23]. In addition, even the same person can change significantly over time. When the nerve fibers that carry sensory information become demyelinated, sensory abnormalities appear (sensory symptoms). On the other hand, when the nerve fibers that carry signals to muscles become demyelinated, movement disorders develop (motor symptoms). Vague symptoms of demyelination in the brain can begin long before the disease is diagnosed, but the most common early symptoms are (i) tingling, numbness, pain, burning, itching, and sometimes loss of touch in the arms, legs, trunk, and face; (ii) weakness or an inability to perform dexterous movements in one leg or hand, which may be accompanied by stiffness in one leg or hand; and (iii) abnormal vision. Vision may become blurry or hazy. Demyelination primarily affects vision when looking straight ahead (central vision), but peripheral vision is less affected. The main symptoms seen during the entire course of MS are vision loss, diplopia, cerebellar ataxia, limb paralysis (monoplegia, paraplegia, and hemiplegia), sensory disturbance, bladder–rectal disorders, gait disturbances, and painful tonic convulsions, which differ depending on the lesion site. Vision problems may also include the following: (iv) Internuclear ophthalmoplegia: when nerve fibers that coordinate the horizontal movements of the eyes when looking left and right are damaged. Because one eye does not turn inward, double vision occurs when people try to look in the opposite direction. In the non-paralyzed eye, the eyeball moves rapidly in one direction and then slowly returns to its original position, causing repeated involuntary movements, known as nystagmus. (v) Optic neuritis (inflammation of the optic nerve): there may be partial loss of vision in one eye and pain when the eye is moved, and problems with walking and balance may also occur.

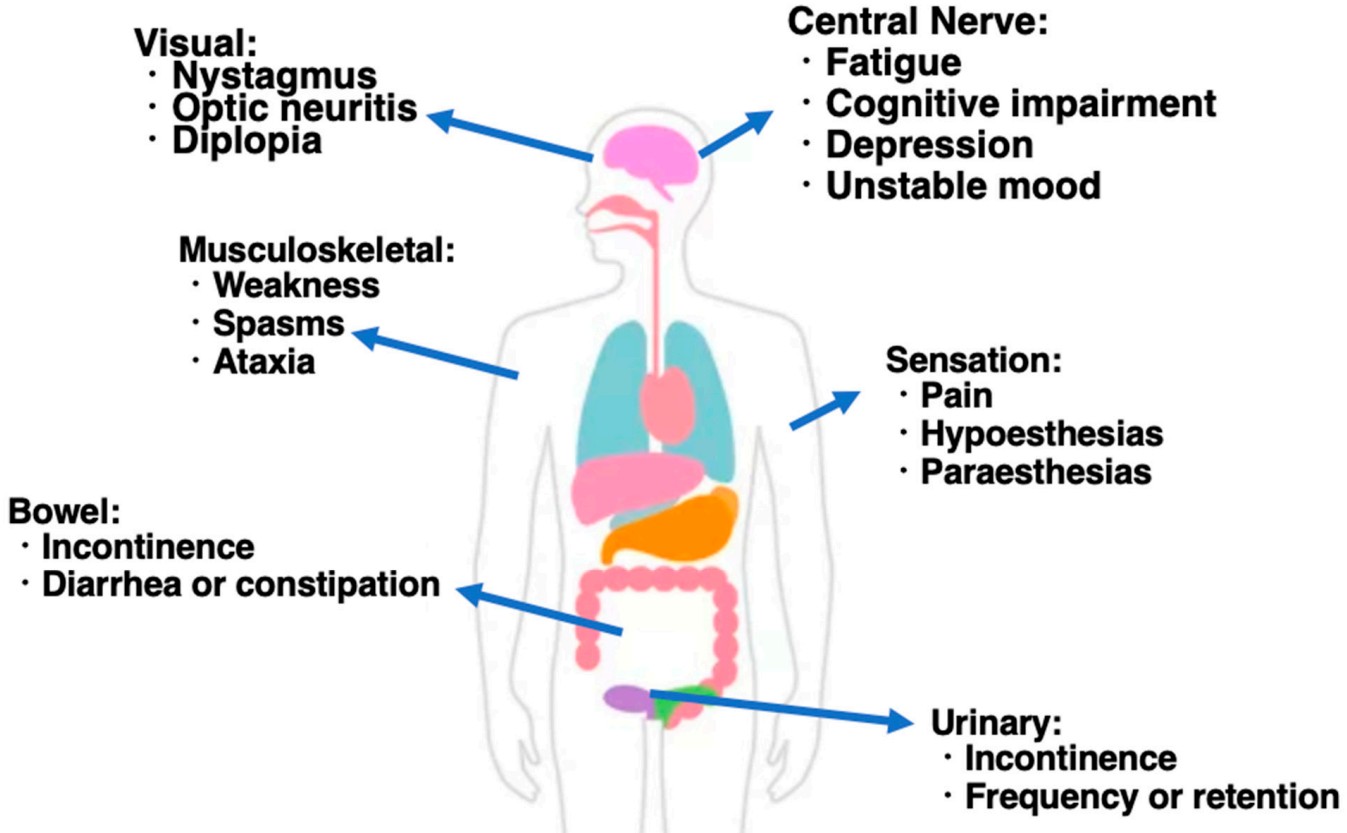

**Figure 1.** Symptoms of MS.

Another symptom characteristic of MS is the Uhthoff sign, in which neurological symptoms worsen with an increase in body temperature and return to normal with a decrease in body temperature [23]. High temperatures, e.g., hot weather, hot baths or showers, and fever, may temporarily worsen symptoms. Additionally, when the posterior part of the spinal cord in the neck is affected and the neck is bent forward, an electric shock or tingling sensation quickly travels down the back to both legs, one arm, and one side of the body—Lhermitte's sign. This sensation is usually fleeting and disappears when the neck is straightened. This unusual sensation often persists while the neck is bent forward.

As a late symptom, movements become shaky and erratic, making it difficult to move the body as MS progresses [24,25]. In addition, partial or complete paralysis may occur. Weak muscles contract involuntarily (spasticity), sometimes causing painful spasms. Weakness and spasticity make walking difficult and eventually impossible even with assistive devices such as walkers. Some people are forced to live in wheelchairs. Being unable to walk can lead to osteoporosis. They may also speak slowly and in a slurred manner. In addition, they may lose control of their emotions, laughing or crying in inappropriate situations. Depression is common and thinking may be mildly disturbed. It often affects the nerves that control urination or defecation. As a result, most patients have problems with urinary control, including frequent urination or strong urge to urinate, urinary incontinence, difficulty initiating urination, inability to completely empty the bladder, and urinary retention. Invoked urine can become a breeding ground for bacteria, making urinary tract infections more likely. Constipation and sometimes fecal incontinence occur, i.e., a loss of control over bowel movements. Rarely, dementia develops after the disease is advanced. More frequent recurrences can lead to severe, sometimes permanent, disability.

## 5. Immunopathogenesis of MS

### 5.1. A Group of B Cells Called Plasmablasts Control Encephalomyelitis via Interleukin-10 (IL-10) Production

In MS, chronic inflammation damages neurons and contributes to disease progression (Figure 2). However, the mechanism of chronic inflammation that occurs in the brain is not fully understood, and the development of therapeutic drugs has not progressed as expected. Early in the onset of EAE, which can be induced by the adjuvant inoculation of mice with specific myelin-constituting peptides such as MOG35-55 as animal models of MS, interleukin-17-producing lymphocytes called Th17 cells induce brain inflammation [26–28]. After the period when symptoms such as tail paralysis, urinary incontinence, and quadriplegia appear, about two weeks after the peptide is inoculated, lymphocytes called "Eomes-positive helper T cells", which express Eomes, a marker of cytotoxic T cells, despite showing helper T cell characteristics, damage nerve cells and cause chronic inflammation in the brain via releasing the cytotoxic protein granzyme B [29,30]. When EAE develops, many lymphocytes and immune cells invade the brain. Among them, B cells and dendritic cells, which have an antigen-presenting function—presenting antigen peptides to T cells—change to produce large amounts of the lactation-stimulating hormone prolactin. Furthermore, B cells and dendritic cells transform T cells that have invaded the brain into "Eomes-positive helper T cells".

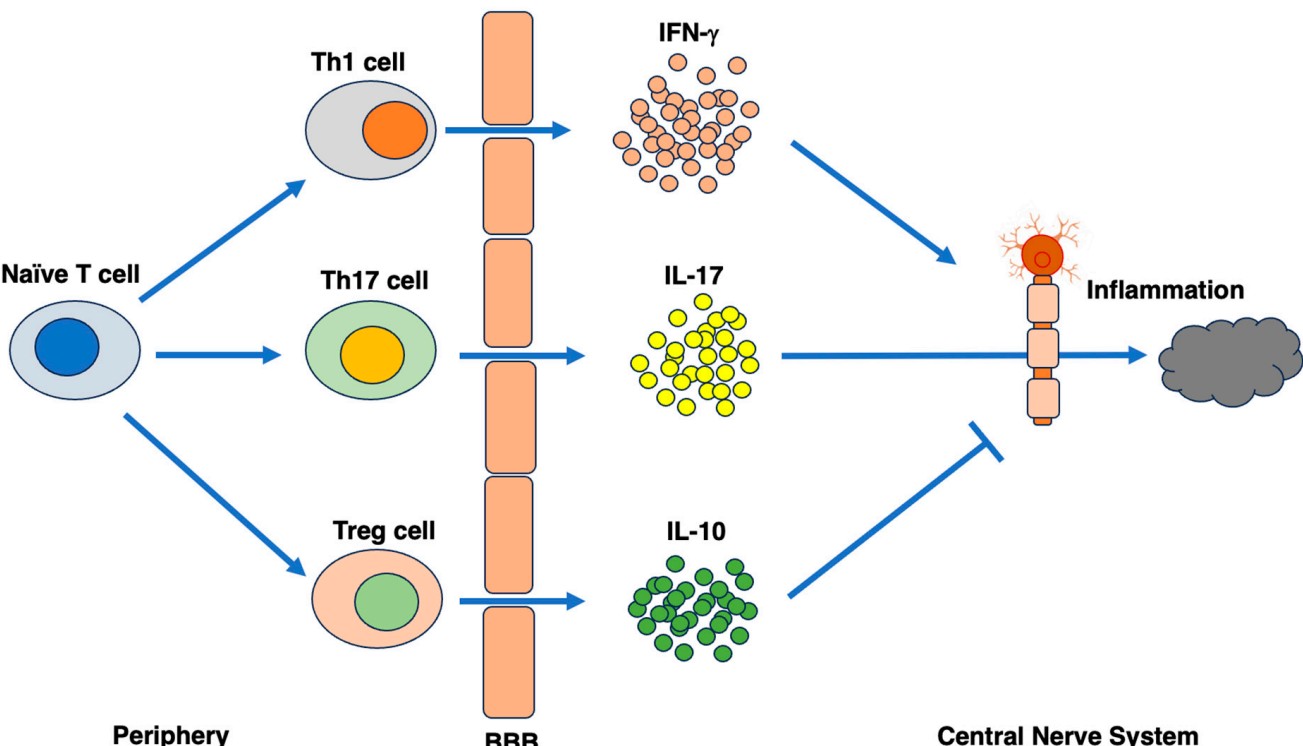

**Figure 2.** T helper cell subsets in MS. The subsets activated in peripheral lymph nodes cross the blood–brain barrier (BBB) and migrate into the central nervous system, where T cells are reactivated by their lineage-defined cytokines, such as interferon-γ (IFN-γ), interleukin-17 (IL-17), and interleukin-10 (IL-10), leading to inflammation and regulation.

B cells play a role in protecting the body from pathogens such as bacteria and viruses by producing antibodies and presenting antigens, but some B cells suppress autoimmune diseases by producing the inhibitory cytokine IL-10. These IL-10-producing B cells, called regulatory B cells, suppress inflammatory responses in experimental mouse models of autoimmune diseases such as inflammatory bowel disease, rheumatoid arthritis, and multiple sclerosis [31–33]. Since CD-1d-expressing CD5-positive B cells in spleen suppress various

autoimmune diseases when administered to mice, and produce IL-10 when stimulated in vitro, these CD-1d-high CD5+ B cells are thought to be the major regulatory B cells. Using the EAE, it became clear that a group of B cells called plasmablasts specifically produce IL-10 and inhibit the function of dendritic cells to suppress the exacerbation of encephalomyelitis [34,35]. However, using EAE models, encephalomyelitis was induced by the subcutaneous injection of peptides of myelin oligodendrocyte glycoprotein (MOG), which is known as a myelin-constituting protein, into IL-10 reporter mice, together with an adjuvant. CD138-positive cells in the regional lymph nodes were found to be the major B cells that produce IL-10 [36,37]. These CD138-positive cells expressed low levels of the transcription factor B-lymphocyte-induced maturation protein (Blimp1), revealing that they were a population of B cells called plasmablasts. The condition of B-cell-specific Blimp1 knockout mice lacking this plasmablast is exacerbated. Based on the above, plasmablasts present in regional lymph nodes act as regulatory B cells that produce IL-10 and suppress the exacerbation of encephalomyelitis. This plasmablast co-localizes mainly with dendritic cells at the boundary between B-cell follicles and T-cell regions. Under the culture supernatant of wild-type plasmablast, the production of inflammatory cytokines such as IL-6 and IL-12a by dendritic cells was decreased, unlike the source of plasmablast culture supernatant lacking IL-10. When naïve T cells expressing MOG-specific T cell receptors and dendritic cells were co-cultured in plasmablast culture supernatant, differentiation into effector T cells was inhibited only in wild-type plasmablast culture supernatants. Thus, plasmablasts inhibit the function of dendritic cells through the production of IL-10, thereby suppressing their differentiation into effector T cells that exacerbate encephalomyelitis.

In B cells stimulated with Toll-like receptors (TLRs), the stimulation of B-cell receptors produces large amounts of IL-10 [38–42]. Furthermore, the $Ca^{2+}$-dependent activation of the transcription factor called nuclear factor of activated T-cells (NFAT) after the stimulation of the B cell receptor is essential to produce IL-10 [43–47]. However, no production of IL-10 is detected in naïve B cells when the B cell receptor is stimulated, despite the activation of NFAT. Therefore, signals from Toll-like receptors (TLRs) are essential to produce IL-10. Only Blimp1-positive B cells produce IL-10 after the stimulation of the B cell receptor. The Blimp1-deficient B cells cannot differentiate into plasma cells, but their early differentiation is induced. In addition, in B cells isolated from interferon regulatory factor 4 (IRF4)-knockout mice, a transcription factor involved in the early differentiation of plasma cells, a low production of IL-10 is observed after the stimulation of B-cell receptors [48–52]. Furthermore, the exacerbation of encephalomyelitis was also observed in mice lacking IRF4 specifically in B cells [53–56]. This suggests that TLR-stimulated B cells express IRF4 and differentiate into plasmablasts, by which they suppress encephalomyelitis by producing IL-10 through the activation of NFAT, dependent on the stimulation of B cell receptors.

In humans, the stimulation of B cells with CpG, a ligand for Toll-like receptor 9 (TLR9), one of the TLRs, and various cytokines such as interferon-a, interleukin-2 (IL-2), and interleukin-6 (IL-6), induces differentiation into plasmablasts [40,57–61]. The in vitro stimulation of B cells isolated from the peripheral blood of healthy individuals with CpG and cytokines induces differentiation into two types of plasmablasts: weakly CD27-positive and strongly CD27-positive. Interestingly, only weakly CD27-positive plasmablasts could produce IL-10 [62–64]. Why do only weakly CD27-positive plasmablasts produce IL-10? Peripheral blood B cells are mainly composed of three distinct B cell populations: naïve mature B cells, naïve immature B cells, and memory B cells. When these B cell populations were isolated from peripheral blood and induced to differentiate into plasmablasts, memory B cells mainly differentiated into plasmablasts that were strongly CD27-positive. On the other hand, naïve mature B cells and naïve immature B cells differentiate into weakly CD27-positive plasmablasts that produce IL-10 [39,65]. Thus, in humans, plasmablasts play an important role in suppressing MS as regulatory B cells that produce IL-10. Thus, plasmablasts not only function as B cells that produce IL-10, which suppresses encephalomyelitis, but are also the major source of B cells that produce IL-10 in humans.

On the other hand, splenic CD1d-expressing CD5-positive B cells suppress encephalomyelitis when transferred to mice despite not producing IL-10 [66,67]. Why can this population of B cells suppress encephalomyelitis? Additionally, is the exacerbation of encephalomyelitis suppressed when plasmablasts that produce IL-10 are transferred to mice? Furthermore, could the decreased production of IL-10 by human plasmablasts be involved in the etiology and pathogenesis of MS? Solving these questions is an important research topic that will lead to the development of new treatments for autoimmune diseases mediated by regulatory B cells.

## 5.2. Programmed Death Receptor-1 (PD-1)-Positive Cells in MS Patients

The immune system is responsible for a complex response that protects host tissues and rapidly destroys invading organisms. T cells play a central role and can be divided into two main groups: helper CD4+ T cells that command other immune cells and killer CD8+ T cells that directly carry out pathogenic cytotoxicity [68,69]. However, the immune response against the host can lead to tissue damage, which is known as an autoimmune disease. Autoimmune pathology is deeply involved in the pathology of MS, an intractable disease in which neuropathy gradually progresses with repeated relapses due to inflammatory disease of the brain, spinal cord, and optic nerve [70,71]. To date, disease-modifying drugs have been developed to control the pathogenicity of CD4-positive T cells. On the other hand, the symptoms of individual patients vary from mild to severe, and some cases are difficult to distinguish from neuromyelitis optica, which requires different treatment methods. CD8-positive T cells, generally known as killer T-cells, infiltrate brain lesions more than CD4-positive T cells and are associated with the degree of neuropathy [72–76]. Recently, regulatory CD8-postitive T cells that suppress autoimmune pathology have attracted attention in animal models of MS [77–81].

Regarding the relationship between the inhibitory immune-receptor PD-1, which is known as a checkpoint of immune regulation, and the pathology of MS among CD8-positive T cells in the peripheral blood, the proportion of PD-1-positive cells is decreased in MS patients compared with healthy controls and is restored by interferon-$\beta$ (IFN$\beta$) treatment [82–86]. In addition, in the cerebrospinal fluid during inflammatory attacks, patients with a high proportion of PD-1+ CD8+ T cells respond better to acute treatment and are more likely to recover to their original level of neuropathy than those with a low proportion [75,87,88]. This PD-1+ CD8+ T cell content correlates with the course of disease recovery. Thus, this cellular subfraction is a suppressor of pathology. Furthermore, in the peripheral blood of patients in MS remission receiving interferon-b treatment, the PD-1+ CD8+ T cell plays a role as an immunoregulatory cell via the immunosuppressive cytokine IL-10 signaling system, suppressing co-receptors such as cytotoxic T-lymphocyte (associated) antigen 4 (CTLA-4), and transcription factor c-Maf, which is involved in immunoregulation [89–93]. Interestingly, two-thirds of the genes co-expressed with PD-1 are potentially under the transcriptional control of c-Maf. In addition, in human CD8-positive T cells forced to express c-Maf, the expression of PD-1 and CTLA-4 is increased, and the production of IL-10 is promoted [89,94–98].

## 5.3. Th17 Cells Regulate the Development of Various Autoimmune Diseases including MS

MS and neuromyelitis optica have an autoimmune pathology in their backgrounds, and the restoration of weakened immune control function can be used as a true disease-modifying treatment. The PD-1+ CD8+ T cells may also serve as indicators that reflect the condition of each patient in other immunological neurological diseases such as neuromyelitis optica. In addition, genes co-expressed with PD-1 include new disease-modifying drug candidates that restore immune control functions, leading to personalized medicine for immune-mediated neurological diseases such as MS. Immunodeficiency occurs when the immune system does not work property, and resistance to pathogens decreases. On the other hand, improper activation can lead to various autoimmune diseases. Among lymphocytes, helper T cells are called the control tower of the immune system because they

regulate the actions of other immune cells. Th17 cells, a type of helper T cell known as the command center of immunity, have been attracting attention in recent years, because they are required for defense against pathogens and promote the development of various autoimmune diseases including MS [99–102]. When naïve CD4+ T cells, which are the source of helper T cells, recognize antigens and survive and proliferate, they differentiate into Th17 cells when stimulated by the cytokines IL-6 and TGF-β [103–106].

However, there are many unclear points regarding how Th17 cells are formed in the body. A transcription factor, JunB (jun B proto-oncogene), has been known to play an important role in physiological functions such as placenta formation, the mechanism of skin barrier maintenance, bone formation, and the regulation of myeloid cell functions by binding to DNA and regulating gene expression [107–110], but what role it plays in helper T cells remains unknown. This JunB deficiency abolishes autoimmune encephalomyelitis, an animal model of MS, and no demyelination or immune cell infiltration, which causes symptoms, was observed [111–114]. By qualitatively and quantitatively regulating JunB to alter the function of Th17 cells, the possibility of developing new treatments for autoimmune diseases was demonstrated [107,115–117]. On the other hand, differentiation into other helper T cells, such as Th1 cells, Th2 cells, and regulatory T cells (Treg), was largely unaffected by JunB deficiency, suggesting that the role of JunB in helper T cells is restricted to Th17 differentiation. JunB also has family proteins with similar amino acid sequences, c-Jun and JunD [118–122]. JunB expression is much higher than that of c-Jun and JunD in Th17 cells and uses a unique segment dissimilar to c-Jun and JunD in its amino acid sequence to induce the expression of genes important for Th17 cell function [116,123].

*5.4. Methodology*

For MS atrophy, since general tests such as blood, urine, and cerebrospinal fluid show no abnormalities, information for diagnosis is collected by head MRI and myocardial scintigraphy. Head MRI shows atrophy of the cerebellum and pons, and atrophy of the basal ganglia, which are characteristic image findings called the "slit sign" and "cross sign" when viewed in a horizontal section [124].

Mouse experimental models of MS, such as EAE induced by myelin oligodendrocyte glycoprotein (MOG) antigens, serve as research tools to investigate the mechanisms underlying autoimmunity and its regulation [125]. In an animal model of MS, EAE, which can be induced by adjuvating mice with specific myelin-constituting peptides, such as MOG35-55, leads to tail paralysis, urinary incontinence, and affected extremities approximately 2 weeks after peptide inoculation, and symptoms persist for a long time. Mice with EAE develop inflammation in the brain and spinal cord, resulting in impaired limb movement. In the early stages of EAE, Th17 cells, lymphocytes that produce IL-17, induce brain inflammation [126,127]. After that period, lymphocytes called "Eomes-positive helper T cells" damage nerve cells and cause chronic inflammation in the brain. When EAE develops, many lymphocytes and immune cells invade the brain. Among them, B cells and dendritic cells, which have an antigen-presenting function, change to produce large amounts of prolactin, a lactation-stimulating hormone. Furthermore, B cells and dendritic cells transform T cells that have invaded the brain into "Eomes-positive helper T cells" through the action of prolactin. Drugs that suppress prolactin secretion also suppress the generation of "Eomes-positive helper T cells" in the brain. Furthermore, this suppresses the chronicity of EAE. Prolactin is known as a lactation-stimulating hormone produced by the pituitary gland. Antigen-presenting cells, B cells, and dendritic cells that invade the brain produce prolactin, which suppresses intracerebral chronic inflammation in EAE. Th17 cells are involved in the early stage of inflammation, and mainly "Eomes-positive helper T cells" are involved in the late stage. These pathology-related molecules are useful tools as diagnostic markers.

## 6. Effects of Pathogenic T Cells Act on Astrocytes in MS via RANKL

MS lesions are widely distributed in the brain, optic nerve, spinal cord, etc., and present with various neurological symptoms [128–134]. This symptom is also characterized by repeated relapses and remissions. As the disease recurs repeatedly, it gradually becomes difficult to recover from neurological dysfunction, and it is difficult to regenerate the central nervous system once it is damaged. This immune system is essentially a system that recognizes and eliminates foreign substances such as pathogenic bacteria and viruses. However, sometimes we misidentify and attack the components within human bodies as foreign substances, which causes inflammation in our own tissues and results in tissue damage. Diseases that cause this phenomenon are called "autoimmune diseases" [135–137]. Among them, MS is one of the diseases caused by the immune system targeting the central nervous system. Nerve fibers in the central nervous system are covered with a membranous structure called the "myelin sheath", which acts as an insulator, and electrical signals flowing through the nerve network are transmitted quickly and accurately. In MS patients, the myelin sheath is destroyed by attacks from the immune system, which impairs neurotransmission and causes neurological symptoms such as visual impairment, sensory impairment, and motor paralysis [138].

The central nervous system has a barrier mechanism called the blood–brain barrier that prevents the influx of harmful substances in the blood. This prevents large molecules such as immune cells, viruses, and toxins from reaching the brain and spinal cord. However, in MS, self-reactive pathogenic T cells penetrate the blood–brain barrier (BBB) and enter the central nervous system, and more immune cells are recruited, causing chronic inflammation [139–143]. Cytokine receptor activator of nuclear factor-kappa B ligand (RANKL), expressed by pathogenic T cells, causes inflammation in the central nervous system [144–148]. In a mouse EAE model in which the RANKL gene is disrupted, both the incidence and progression of the disease are strongly suppressed [149,150]. Furthermore, in this mouse model, inflammatory cells such as pathogenic T cells and macrophages cannot enter the central nervous tissue, and both inflammation and myelination in the central nervous tissue are strongly suppressed. Astrocytes in the central nervous system express RANK receptors for RANKL, and pathogenic T cells act on astrocytes via RANKL [148,150]. In addition, astrocytes stimulated by RANKL produce large amounts of chemokines, which are factors that promote cell migration [151,152]. As a result, many pathogenic T cells and inflammatory cells are attracted to the central nervous system, causing chronic inflammation. In fact, EAE development is suppressed even in mice deficient in RANK receptors, but only in astrocytes [149–152]. Furthermore, the oral administration of a small-molecule inhibitor of RANKL to mice reduces the incidence of EAE [149,153]. Based on these findings, it is expected that RANKL-targeted therapy will exert a strong effect on MS. The mainstay of treatment for MS has been steroid preparations and interferon therapy, which broadly suppress immune responses [154,155]. However, in recent years, new drugs such as fingolimod and natalizumab, which suppress T cell trafficking, have been developed and are attracting attention [156,157]. A therapeutic method targeting RANKL is expected to bring about a further leap forward in this new therapeutic approach, targeting T cell trafficking.

## 7. Discussion

MS is broadly divided into relapsing remitting MS (RRMS), in which relapses and remissions follow a natural course, and primary progressive MS (PPMS), in which a chronic progressive course develops from the onset of the disease [158,159]. In the early stages of RRMS, sequelae of recurrence often do not remain. However, the aftereffects gradually remain after each recurrence. PPMS has a male-to-female ratio of 1:1 and no female dominance like RRMS. PPMS also has an age of onset 10 years later than RRMS but progresses more rapidly. Movement disorders such as spastic paraplegia and cerebellar ataxia are the main symptoms and progress slowly. Current disease-modifying drugs are effective in reducing relapse in RRMS but are completely ineffective in slowing the progression of disability in PPMS.

On the other hand, genome-wide association analysis showed no significant difference in disease susceptibility genes between RRMS and PPMS, supporting the idea that they are different phenotypes of the same disease. In MS, high latitude and the associated low sun exposure, Epstein–Barr (EB) virus infection, vitamin D deficiency, smoking, and month of birth have been shown to be environmental risks. Furthermore, the MS concordance rate was higher in dizygotic twins (5.4%) than in siblings (2.9%), suggesting the importance of a common fetal environment in dizygotic twins [160,161]. In the northern hemisphere, MS is more common in those born in April and May and less common in those born in October and November, supporting the importance of the fetal environment [162–164]. One explanation for this is that maternal and fetal vitamin D levels are low in winter, because vitamin D induces neuroprotective helper T cell 2 (Th2) cells and suppresses helper T cell 1 (Th1). In addition, migrating before the age of 15 changes the prevalence of MS at the destination, suggesting that the environment from birth to puberty is important. Furthermore, the onset of a recurrence of MS increases two to three times more during non-specific infection than during non-infection, supporting the importance of infectious factors as triggers [165,166].

MS is a chronic inflammatory demyelinating disease of the central nervous system caused by an autoimmune mechanism, in which the pathophysiological mechanism at the molecular level can be divided into three stages; the first step is the process of establishing an autoimmune response against myelin constituent proteins of the central nervous system, including MBP [165–167]. This occurs outside the central nervous system and in the peripheral lymphoid tissue that exists outside the BBB, and the molecular homology between infectious agents such as EB virus and MBP is important, since Th cells bearing the CD4 antigen are the main players in the autoimmune response in MS. However, for the Th response to exert its function, antigen-presenting cells (APC) such as macrophages and dendritic cells must be stimulated by the presentation of modified peptide antigens. When antigen molecules on HLA class II molecules present on the surface of APCs fit into their own T cell receptor structures, the Th cells activate and differentiate into various effector T cells depending on their surroundings, e.g., cytokine environment [168,169]. When stimulated and induced to differentiate, T cells are not presented with antigens in a strict lock-and-key relationship. It is known that when the positions and positive or negative charge states of some amino acid residues match a certain pattern, they are recognized as target antigens. In addition, to activate the Th response, at the same time as receiving an appropriate antigen presentation from APC, it is essential that co-stimulatory signals enter through ligands for the co-stimulatory molecule CD28 present on the Th cell side [170]. Attempts have been made to paralyze or modify the function of MBP-reactive T cells to render them harmless.

The second stage is passage through the BBB, in which T cells that have received antigen presentation in peripheral lymph nodes or spleen are activated and circulate in the peripheral blood; the activated T cells can cross the BBB relatively easily. After entering the central nervous system, they are presented by the microglia with antigens that function as APCs, but unless they are presented with the originally determined target antigens, they migrate through the tissue and then exit the central nervous system again [171–174]. In this way, the central nervous system is constantly monitored for the presence of infectious antigens and tumor cells by patrolling the interior of the central nervous system with a certain percentage of activated T cells present in the peripheral blood. The involvement of adhesion molecules is essential for crossing the BBB, but in general, leukocytes, including lymphocytes, cannot easily exit the blood vessel when they are in a fast flow of blood. For example, L-selectin expressed on the surface of T cells begins with capture into the vascular endothelium, followed by rolling on the endothelium surface [175]. Furthermore, the binding of chemokines present on the surface of the endothelium with chemokine receptors on T cells leads to adhesion molecule ligands being activated and then terminated by binding to endothelial adhesion molecules. As adhesion molecule ligands are used by T cells to cross the BBB, alphaLbeta3 integrin, leukocyte function-

associated molecule-1 (LFA-1), and very late antigen (VLA)-4 (alpha4beta1 integrin) are important [176]. They firmly adhere cell bodies onto endothelial cells by binding the former and the latter with the intercellular adhesion molecules (ICAM)-1 on the endothelium and the vascular cell adhesion molecule (VCAM)-1, respectively [177]. After that, T cells pass between endothelial cells or within endothelial cell bodies, and then cross the BBB, lined by the basement membrane and astrocyte foot processes, to enter the central nervous system.

There is a monoclonal antibody (natalizumab) against human VLA-4 (CD49d molecule) that targets alpha4beta1 integrin, which is expressed in many T cells involved in tissue inflammation in EAE, which is a treatment aimed at blocking the entry of proinflammatory T cells into the central nervous system [178]. On the other hand, sphingosine 1-phosphate receptors expressed on the surface of T cells play an important role in the emergence of T cells from the medullary sinuses of lymph nodes into the bloodstream via the effect lymphatic vessels. Fingolimod, which functionally inhibits this receptor, markedly reduces the number of circulating central memory T cells, resulting in an indirect inhibitory effect on central nervous system invasion [179]. Both drugs show sufficient efficacy in preventing the recurrence of MS, but it is necessary to be aware of the risks of these treatments as they may affect the constant patrol of the brain by T cells.

The third step is the immune and inflammatory response within the central nervous system. Autoreactive Th that has invaded the central nervous system is reactivated when it is presented with some of the components that make up myelin from APC microglia, resulting in the release of proinflammatory cytokines such as IFNb and TNF-a [180]. These cytokines promote the expression of HLA class II antigens in microglia, enhancing their ability to present antigens, and IL-2 produced by microglia promotes the induction of Th1 [5,168]. In addition, endothelial cells become activated and increase the expression of adhesion molecules, resulting in the further amplification of the autoimmune response and the recruitment of macrophages from the peripheral blood into the central nervous system [181–183]. Th1 cells that produce IFNβ and Th17 cells that produce IL-17 are known exacerbating factors that promote this series of processes [184–187]. In terms of improving factors that suppress MS pathology, there are Th2, which produces the anti-inflammatory cytokines IL-4 and IL-13; Th3, which produces TGF-beta; IL-10-producing Tr1; and regulatory T cells (Treg), which express a large amount of the CD25 antigen on the cell surface. All of these factors are immunocompetent cells belonging to the CD4-positive helper T-cell lineage [141,188,189]. IFNβ is known to bias the response toward Th2 and increase Tregs [189,190]. On the other hand, many CD8 cells perivascularly infiltrate the demyelinating plaques of MS [80,191,192]. The process that ultimately leads to demyelination is the mechanism by which TNF-a and nitric oxide released from Th1 cells and macrophages damage myelin and oligodendrocytes, and autoantibodies mainly composed of IgG bind to myelin. The binding of complement to the Fc receptor of the antibody activates the complement cascade, leading to phagocytosis by macrophages [193].

Additionally, once myelin is destroyed, components other than the original target antigens are exposed, and the number of new target antigens increases, gradually expanding the scope of the autoimmune response [73,194]. Therefore, therapeutic intervention in the early stage of MS can effectively suppress disease activity. Thus, MS is a disease that causes multiple inflammatory lesions in the central nervous system. Clinically, relapses and remissions repeat over many years, but neurodegeneration progresses gradually. On the other hand, there are MS cases that respond to disease-modifying drugs for a long period of time [195,196]. For example, patients who are successfully treated with IFNβ have a stable course. Therefore, it is important to elucidate the difference between progressive MS and benign MS, but it is difficult to observe tissue changes over time, and some kind of template model is necessary to approach the core of the changing immune pathology. The dysfunction of inhibitory receptors expressed on T cells is known to be the core of MS immunopathology and may be the cause of chronic persistent inflammation [84,197]. Originally, T cells that received inhibitory signals from tissues should become exhausted T-cell-like cells. However, in MS, the differentiation process is some-

how imperfect, and chronic inflammation persists, potentially leading to neuropathy. In fact, in the group of MS patients with good steroid treatment responsiveness, the number of PD-1-positive T cells in the cerebrospinal fluid was high, and conversely, the PD-1-positive rate decreased in the group with residual symptoms of neuropathy even after treatment [85]. Thus, MS and neuromyelitis optica both have autoimmune conditions in their backgrounds, and the restoration of weakened immune regulatory functions may be a true disease-modifying treatment.

MS is caused by "demyelination" in the central nervous system, that is, in the brain and spinal cord [1,2]. Various symptoms such as visual impairment, numbness, motor paralysis, dizziness, and dysuria are observed. In the early stages of the disease, the symptoms temporarily and spontaneously improve, but the recovery gradually deteriorates, and aftereffects gradually appear. In addition, brain atrophy and cognitive impairment worsen from an early stage even without the recurrence of symptoms. Chronic "inflammation" is involved in the background of "demyelination" [198]. It is an intractable disease that cannot be completely cured. Currently, various therapeutic drugs, including disease-modifying drugs, are available, and early diagnosis and appropriate treatment selection may significantly reduce long-term disease progression. The central nervous system is composed of neurons, oligodendrocytes, astrocytes, microglia, and cerebral vessels. In MS, oligodendrocytes that wrap nerve cells are damaged by "inflammation". In the early stage of the disease, the brain has reserve power, and even if "inflammation" is strongly observed, there may be no symptoms, or the spontaneous remission of symptoms may be observed. However, reserve power declines with age, and recovery gradually deteriorates, leaving aftereffects. In addition, because of "inflammation", brain atrophy progresses from the early stages of the disease, leading to cognitive impairment. In MS, therefore, it is important to tightly manage the "inflammation" or disease activity. Since disease activity is strongest in the early stages of the disease and then gradually declines, it is necessary to select an appropriate treatment from the early stages of the disease, even if the patient is asymptomatic. In diagnosing MS, it is important to "exclude other diseases", in addition to "spatial and temporal recurrence in different sites". Head MRI examination, blood, and cerebrospinal fluid examination are necessary for diagnosis, and it is necessary to make a firm diagnosis before intervening with treatment.

## 8. Conclusions

As the source of the immunopathogenesis of MS, a group of B cells called plasmablasts control encephalomyelitis via IL-10 production. These IL-10-producing B cells, called regulatory B cells, suppress inflammatory responses in experimental mouse models of autoimmune diseases, including MS. Among CD8-positive T cells in the peripheral blood, the proportion of PD-1-positive cells is decreased in MS patients compared with healthy controls. The PD-1+ CD8+ T cells may also serve as indicators that reflect the condition of each patient in immunological neurological diseases such as MS. In addition, the gene cluster co-expressed with PD-1 contains new disease-modifying drug candidates that restore immunoregulatory function, which is expected to lead to personalized medicine for immunological neurological diseases such as MS.

**Author Contributions:** Writing—review and editing, Y.M.; supervision, R.Y.; funding acquisition, Y.M. All authors have read and agreed to the published version of the manuscript.

**Funding:** This research received no external funding.

**Conflicts of Interest:** The authors declare no conflict of interest.

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
