# Peer review of "Unraveling the Immunopathogenesis of Multiple Sclerosis: The Dynamic Dance of Plasmablasts and Pathogenic T Cells"

_biologics, doi:10.3390/biologics3030013_

Round 1
Reviewer 1 Report
This review summarizes current knowledge about immunopathogenesis of Multiple Sclerosis. In my opinion, the article is well written, well documented and contributes significantly to the understanding of immune responses generated by the disease in humans or experimental models.
There are few points, however, which I would like to remarck or comment.
- First, the abstract should mention, even briefly, the objective of the article, not only summarizing the main points.
- I enjoyed reading the introduction, but I expected some posterior mention to the interesting anti-AQP4 antibodies, regarding their probable function in immunopathogenesis (the main point of the review), or related to symptoms. They appear to be a central part of the article as they are mentioned in abstract, as keywords and in the conclusion; however in the main part of the text they are absent.
- In the Symptoms section (line numbers would be helpful), please add references representing the whole paragraph information. For example, it is not clear if references 17-19 support the information stated below the reference in the text. This situation repeats along the text in several parts.
- In Immunopathogenesis section, there are several acronyms which are defined much after the first use (for examples: , IFNb, MOG, IL-10 -and others-, TLR). Please consider reorganizing the text and revise that they are defined the first time and then only use the acronym.
- The figures are not mentioned in the text.
Author Response
This review summarizes current knowledge about immunopathogenesis of Multiple Sclerosis. In my opinion, the article is well written, well documented and contributes significantly to the understanding of immune responses generated by the disease in humans or experimental models.
There are few points, however, which I would like to remarck or comment.
- First, the abstract should mention, even briefly, the objective of the article, not only summarizing the main points.
According to reviewer’s comment, we added some sentences in abstract section.
- I enjoyed reading the introduction, but I expected some posterior mention to the interesting anti-AQP4 antibodies, regarding their probable function in immunopathogenesis (the main point of the review), or related to symptoms. They appear to be a central part of the article as they are mentioned in abstract, as keywords and in the conclusion; however in the main part of the text they are absent.
According to reviewer’s comment, we added some sentences in second section, 2. Pathogenesis of MS.
- In the Symptoms section (line numbers would be helpful), please add references representing the whole paragraph information. For example, it is not clear if references 17-19 support the information stated below the reference in the text. This situation repeats along the text in several parts.
According to reviewer’s comment, we added sub-titles abd line numbers.
- In Immunopathogenesis section, there are several acronyms which are defined much after the first use (for examples: , IFNb, MOG, IL-10 -and others-, TLR). Please consider reorganizing the text and revise that they are defined the first time and then only use the acronym.
According to reviewer’s comment, we corrected and reorganized the acronyms.
- The figures are not mentioned in the text.
According to reviewer’s comment, we added figures in the text.
Reviewer 2 Report
The topic is of interest, however this review has at the moment some limitations that need to be fixed:
-in the abstract and the conclusion, I would suggest removing the sentence "Involvement of serum anti-aquaporin 4 (AQP4) antibodies in pathogenesis of MS suggests various clinical patterns of MS with neuromyelitis optica spectrum disorders." NMOSD and MS are 2 completely different disease.
-the abstract is not well representative of what the reader can find in the review
-What is the aim and the methodology of the review? What does this review add to the already published papers about the topic? How did you select the info in the review? Please address this matter.
-the title is very specific but it is not until page 5 that we begin to read about MS pathopisiology...perhaps the part concerning MS simptom is redundand?
-the images are very good.
The topic is of interest, however this review has at the moment some limitations that need to be fixed:
-in the abstract and the conclusion, I would suggest removing the sentence "Involvement of serum anti-aquaporin 4 (AQP4) antibodies in pathogenesis of MS suggests various clinical patterns of MS with neuromyelitis optica spectrum disorders." NMOSD and MS are 2 completely different disease.
-the abstract is not well representative of what the reader can find in the review
-What is the aim and the methodology of the review? What does this review add to the already published papers about the topic? How did you select the info in the review?
-the title is very specific but it is not until page 5 that we begin to read about MS pathopisiology...perhaps the part concerning MS simptom is redundand?
-the images are very good.
Author Response
The topic is of interest, however this review has at the moment some limitations that need to be fixed:
-in the abstract and the conclusion, I would suggest removing the sentence "Involvement of serum anti-aquaporin 4 (AQP4) antibodies in pathogenesis of MS suggests various clinical patterns of MS with neuromyelitis optica spectrum disorders." NMOSD and MS are 2 completely different disease.
According to reviewer’s comment, we deleted the sentence that "Involvement of serum anti-aquaporin 4 (AQP4) antibodies in pathogenesis of MS suggests various clinical patterns of MS with neuromyelitis optica spectrum disorders."
-the abstract is not well representative of what the reader can find in the review
According to reviewer’s comment, we added some sentences in abstract section.
-What is the aim and the methodology of the review? What does this review add to the already published papers about the topic? How did you select the info in the review? Please address this matter.
According to reviewer’s comment, we added some sentences in some section for addressing advises.
-the title is very specific but it is not until page 5 that we begin to read about MS pathopisiology...perhaps the part concerning MS simptom is redundand?
According to reviewer’s comment, we corrected title into "Unraveling the Immunopathogenesis of Multiple Sclerosis: The Dynamic Dance of Plasmablasts and Pathogenic T Cells".
-the images are very good.
Reviewer 3 Report
The review article by Matsuzaka and Yashiro on "Immunopathogenesis of Multiple Sclerosis with Plasmablasts and Pathogenic T Cells" is a well written and scientifically valid article.
I do not have much criticism to make on this article since it has covered relevant features. But I strongly feel the authors should dedicate a section where they explicitly discuss why and how this review article is important for the field. Otherwise it may seem like just another review article in the ocean of many.
The potential treatment therapies targeting these specific pathways mentioned in this review need to be summarized in a table.
Also, the font size used throughout this article are inconsistent in some lines. There are few minor spelling mistakes too. I would request the authors to give it a thorough check.
Except these, I have no major comments to make.
Author Response
The review article by Matsuzaka and Yashiro on "Immunopathogenesis of Multiple Sclerosis with Plasmablasts and Pathogenic T Cells" is a well written and scientifically valid article.
I do not have much criticism to make on this article since it has covered relevant features. But I strongly feel the authors should dedicate a section where they explicitly discuss why and how this review article is important for the field. Otherwise it may seem like just another review article in the ocean of many.
According to reviewer’s comment, we added discussion section for explaining the significance of this review article in this field.
The potential treatment therapies targeting these specific pathways mentioned in this review need to be summarized in a table.
According to reviewer’s comment, we summarized treatment therapies as Table 1.
Also, the font size used throughout this article are inconsistent in some lines. There are few minor spelling mistakes too. I would request the authors to give it a thorough check.
According to reviewer’s comment, we corrected the font size and minor spelling mistakes.
Except these, I have no major comments to make.
Reviewer 4 Report
24 July 2023
Manuscript ID: biologics-2487904
Type: Review
Title: “Immunopathogenesis of Multiple Sclerosis with Plasmablasts and Pathogenic T Cells” by Matsuzaka Y & Yashiro R, submitted to Biologics
Dear Authors,
Current challenges in multiple sclerosis (MS) include the development of more effective therapies that target the underlying immunopathogenesis of the disease, the identification of biomarkers for early diagnosis and monitoring of disease progression, and the creation of personalized treatment strategies based on the unique patient characteristics. In the present review article, entitled "Immunopathogenesis of Multiple Sclerosis with Plasmablasts and Pathogenic T Cells," Matsuzaka and Yashiro discuss the role of plasmablasts and pathogenic T cells in the development of the disease, and highlighting the current challenges and future directions in the field.
The manuscript's main strength is that it addresses a timely and fascinating topic, providing a comprehensive review of the immunopathogenesis of MS and discussing the latest research on the role of plasmablasts and pathogenic T cells in the development of the disease. The authors also highlight the need for more effective therapies, the identification of biomarkers for early diagnosis and monitoring of disease progression, and the development of personalized treatment strategies based on individual patient characteristics.
In general, I think the idea of this article is really interesting, and the authors’ fascinating observations on this timely topic may be of interest to the readers of Biologics. However, some comments, as well as some crucial evidence that should be included to support the author’s argumentation, needed to be addressed to improve the quality of the manuscript, its adequacy, and its readability prior to its publication in the present form. My overall judgment is to publish this paper after the authors have carefully considered my suggestions below.
Please consider the following comments:
1. Title: This is the most important section of the manuscript. Please present a concise and self-explanatory title stating the most important message of this review. Suggestions: "Unraveling the Immunopathogenesis of Multiple Sclerosis: The Dynamic Dance of Plasmablasts and Pathogenic T Cells"; "Unveiling the Puzzle: Plasmablasts and Pathogenic T Cells in the Immunopathogenesis of Multiple Sclerosis"[1–3].
2. Abstract: I suggest the authors present the background, a short summary, and a conclusion in proportional order within the 200 words, according to the journal’s guidelines [4]. The general background (one to two sentences), the specific background (two to three sentences), and the current issue covered by this review (one sentence) should all be included in the background before moving on to the objectives. I would like the author to provide background information, a problem statement, and their reasoning for branching off in this subsection. The brief review section concludes with a phrase that places this subsection in a broader context. The conclusion should begin with one sentence that summarizes the main message using words like "Here we highlight." The authors should describe the potential and the advancement this study has made in the field in the first sentence of the conclusion, followed by two to three sentences that provide a broader perspective that is easily understood by a scientist from any discipline [5–8].
3. Keywords: Please list ten keywords chosen from Medical Subject Headings (MeSH) and use as many as possible in the title and in the first two sentences of the abstract [9].
4. A graphical abstract that will visually summarize the main findings of the manuscript is highly recommended.
5. Introduction: I would like the authors to reorganize this section with about 1000 words and several paragraphs, introduce information on the key study constructs that should be understood by readers in any discipline, and make it persuasive enough to advance the primary goal of the author's recent research and the particular goal the author has intended by this review. I would like to suggest that the authors present the introduction beginning with the overall context, moving on to the specific context, and concluding with the current problem addressed in this review before moving on to the objectives. Those key structures ought to be set up logically and coherently [10] I also recommend that the authors provide the rationale for presenting subsequent sections in order to assist the reader in navigating the document.
6. In this regard, I believe that the following works, but not limited to, may enhance the value of this manuscript [11–17].
7. Discussion: I would like the authors to present the independent disussion section by opening with an introductory paragraph and followed by the summary of the previous sections. Then, I expect the authors to develop arguments clarifying the potential of this study as an extension of the previous work, the implication of the findings of this study, how this study could facilitate future research, the ultimate goal, the challenge, the knowledge and technology necessary to achieve this goal, the statement about this field in general, and finally the importance of this line of research. It is particularly important to present the limits, merit, and potential translation of this study to clinical practice [18,19].
8. Conclusion: I think that presenting the conclusion would benefit from a single paragraph presenting some thoughtful as well as in-depth considerations by the authors as experts to convey the take-home message. The authors should make an effort to explain the theoretical implications as well as the translational application of their research. I believe that it would be necessary to discuss theoretical and methodological avenues in need of refinement as well as suggestions for a path forward in understanding the importance of this study.
9. References: Please follow the guidelines of the journal [4]. The journal names should be italicized.
Overall, the manuscript contains two figures, no tables, and 153 references. I believe that the manuscript may have important value in presenting a valuable resource for anyone interested in the immunopathogenesis of MS. It provides a detailed overview of the latest research on the role of plasmablasts and pathogenic T cells in the development of the disease and highlights the current challenges and future directions in the field. I hope that, after these careful revisions, the manuscript can meet the journal’s high standards for publication. I am available for a new round of revisions to this article. I hope that, after these careful revisions, this paper can meet the journal’s high standards for publication. I am available for a new round of revisions to this article.
I declare no conflict of interest regarding this manuscript.
Best regards,
Reviewer
References:
- https://plos.org/resource/how-to-write-a-great-title/
- https://www.nature.com/nature-index/news-blog/how-to-write-a-good-research-science-academic-paper-title
- https://www.indeed.com/career-advice/career-development/catchy-title
- https://www.mdpi.com/journal/biologics/instructions
- https://www.scribbr.com/dissertation/abstract/
- https://writing.wisc.edu/handbook/assignments/writing-an-abstract-for-your-research-paper/
24 July 2023
Manuscript ID: biologics-2487904
Type: Review
Title: “Immunopathogenesis of Multiple Sclerosis with Plasmablasts and Pathogenic T Cells” by Matsuzaka Y & Yashiro R, submitted to Biologics
Dear Authors,
Based on the English proficiency assessment, it is noted that minor editing of the English language is required. While the overall communication is clear and understandable, there are some areas that could benefit from slight improvements in grammar, syntax, and word choice. Attention to detail, such as refining sentence structure and ensuring proper tense usage, will enhance the overall coherence and fluency of the written work. With some minor editing adjustments, the English language proficiency can be further enhanced.
Best regards,
Reviewer
Author Response
Manuscript ID: biologics-2487904
Type: Review
Title: “Immunopathogenesis of Multiple Sclerosis with Plasmablasts and Pathogenic T Cells” by Matsuzaka Y & Yashiro R, submitted to Biologics
Dear Authors,
Current challenges in multiple sclerosis (MS) include the development of more effective therapies that target the underlying immunopathogenesis of the disease, the identification of biomarkers for early diagnosis and monitoring of disease progression, and the creation of personalized treatment strategies based on the unique patient characteristics. In the present review article, entitled "Immunopathogenesis of Multiple Sclerosis with Plasmablasts and Pathogenic T Cells," Matsuzaka and Yashiro discuss the role of plasmablasts and pathogenic T cells in the development of the disease, and highlighting the current challenges and future directions in the field.
The manuscript's main strength is that it addresses a timely and fascinating topic, providing a comprehensive review of the immunopathogenesis of MS and discussing the latest research on the role of plasmablasts and pathogenic T cells in the development of the disease. The authors also highlight the need for more effective therapies, the identification of biomarkers for early diagnosis and monitoring of disease progression, and the development of personalized treatment strategies based on individual patient characteristics.
In general, I think the idea of this article is really interesting, and the authors’ fascinating observations on this timely topic may be of interest to the readers of Biologics. However, some comments, as well as some crucial evidence that should be included to support the author’s argumentation, needed to be addressed to improve the quality of the manuscript, its adequacy, and its readability prior to its publication in the present form. My overall judgment is to publish this paper after the authors have carefully considered my suggestions below.
Please consider the following comments:
- Title: This is the most important section of the manuscript. Please present a concise and self-explanatory title stating the most important message of this review. Suggestions: "Unraveling the Immunopathogenesis of Multiple Sclerosis: The Dynamic Dance of Plasmablasts and Pathogenic T Cells"; "Unveiling the Puzzle: Plasmablasts and Pathogenic T Cells in the Immunopathogenesis of Multiple Sclerosis"[1–3].
According to reviewer’s comment, we corrected title into "Unraveling the Immunopathogenesis of Multiple Sclerosis: The Dynamic Dance of Plasmablasts and Pathogenic T Cells".
- Abstract: I suggest the authors present the background, a short summary, and a conclusion in proportional order within the 200 words, according to the journal’s guidelines [4].The general background (one to two sentences), the specific background (two to three sentences), and the current issue covered by this review (one sentence) should all be included in the background before moving on to the objectives. I would like the author to provide background information, a problem statement, and their reasoning for branching off in this subsection. The brief review section concludes with a phrase that places this subsection in a broader context. The conclusion should begin with one sentence that summarizes the main message using words like "Here we highlight." The authors should describe the potential and the advancement this study has made in the field in the first sentence of the conclusion, followed by two to three sentences that provide a broader perspective that is easily understood by a scientist from any discipline [5–8].
According to reviewer’s comment, we corrected Abstract section.
- Keywords: Please list ten keywords chosen from Medical Subject Headings (MeSH) and use as many as possible in the title and in the first two sentences of the abstract [9].
According to reviewer’s comment, we corrected keywords by choosing from MeSH.
- A graphical abstract that will visually summarize the main findings of the manuscript is highly recommended.
According to reviewer’s comment, we attached a graphical abstract.
- Introduction: I would like the authors to reorganize this section with about 1000 words and several paragraphs, introduce information on the key study constructs that should be understood by readers in any discipline, and make it persuasive enough to advance the primary goal of the author's recent research and the particular goal the author has intended by this review. I would like to suggest that the authors present the introduction beginning with the overall context, moving on to the specific context, and concluding with the current problem addressed in this review before moving on to the objectives. Those key structures ought to be set up logically and coherently [10] I also recommend that the authors provide the rationale for presenting subsequent sections in order to assist the reader in navigating the document.
According to reviewer’s comment, we corrected Introduction section by reorganizing and adding the current problem.
- In this regard, I believe that the following works, but not limited to, may enhance the value of this manuscript [11–17].
- Discussion: I would like the authors to present the independent disussion section by opening with an introductory paragraph and followed by the summary of the previous sections. Then, I expect the authors to develop arguments clarifying the potential of this study as an extension of the previous work, the implication of the findings of this study, how this study could facilitate future research, the ultimate goal, the challenge, the knowledge and technology necessary to achieve this goal, the statement about this field in general, and finally the importance of this line of research. It is particularly important to present the limits, merit, and potential translation of this study to clinical practice [18,19].
According to reviewer’s comment, we added discussion section.
- Conclusion: I think that presenting the conclusion would benefit from a single paragraph presenting some thoughtful as well as in-depth considerations by the authors as experts to convey the take-home message. The authors should make an effort to explain the theoretical implications as well as the translational application of their research. I believe that it would be necessary to discuss theoretical and methodological avenues in need of refinement as well as suggestions for a path forward in understanding the importance of this study.
According to reviewer’s comment, we added some sentences in conclusion section.
- References: Please follow the guidelines of the journal [4]. The journal names should be italicized.
According to reviewer’s comment, we corrected references by italicizing.
Overall, the manuscript contains two figures, no tables, and 153 references. I believe that the manuscript may have important value in presenting a valuable resource for anyone interested in the immunopathogenesis of MS. It provides a detailed overview of the latest research on the role of plasmablasts and pathogenic T cells in the development of the disease and highlights the current challenges and future directions in the field. I hope that, after these careful revisions, the manuscript can meet the journal’s high standards for publication. I am available for a new round of revisions to this article. I hope that, after these careful revisions, this paper can meet the journal’s high standards for publication. I am available for a new round of revisions to this article.
I declare no conflict of interest regarding this manuscript.
Best regards,
Round 2
Reviewer 2 Report
“According to reviewer’s comment, we added some sentences in some section for addressing advises”. Unfortunately, I don’t see in the revised manuscript any methodology/methods section… As previously stated, this needs to be fixed.
Moreover, what does this review add to the already-published one? You need to identify the gap in knowledge and express it in the methods section.
Plus, the discussion section was highly implemented, which is nice, but usually the discussion describes what information the overview presents to the reader and discusses it critically. What is the opinion of the authors? What is the gap in knowledge? What research is needed in the future?
Thank you
Author Response
“According to reviewer’s comment, we added some sentences in some section for addressing advises”. Unfortunately, I don’t see in the revised manuscript any methodology/methods section… As previously stated, this needs to be fixed.
According to reviewer’s comment, we added 5-4. Methodology/Methods section on line 392 to 420, in page 14 to 15 to clear the analytical method of animal model of MS, in which critical factors in analytical terms are summurized. If it were insufficient as Methodology/Methods section, please let me know.
Moreover, what does this review add to the already-published one? You need to identify the gap in knowledge and express it in the methods section.
According to reviewer’s comment, we added some sentences in introduction section on line 62 to 119, in page 2 to 4 to clear the gap between the published reports and unresolved issues.
Plus, the discussion section was highly implemented, which is nice, but usually the discussion describes what information the overview presents to the reader and discusses it critically. What is the opinion of the authors? What is the gap in knowledge? What research is needed in the future?
According to reviewer’s comment, we added some sentences in discussion section on line 596 to 620, in page 20 to 21 to clear the gap, future, opinion of this review that is close-up the ivolvement of inflammation in MS.
Thank you
Reviewer 4 Report
16 August 2023
Manuscript ID: biologics-2487904
Type: Review
Title: “Immunopathogenesis of Multiple Sclerosis with Plasmablasts and Pathogenic T Cells” by Matsuzaka Y & Yashiro R, submitted to Biologics
Dear Authors,
I am glad to see that the authors have made an effort to revise the manuscript. Regarding my review report, however, only partial revisions have been made. Prior to publication, I kindly request that the authors carefully consider my comments from the previous report. Here, I highlight unrevised sections with quotation marks "..." to assist the authors in revising the manuscript to meet the journal's high standards. Also, I anticipate the authors preparing a detailed rebuttal to my remarks.
Comments:
1. Abstract: Please make an effort to present main components proportionally: I suggest the authors present “the background”, “a short summary”, and “a conclusion” in proportional order within the 200 words, according to the journal’s guidelines [4]. The general background (one to two sentences), the specific background (two to three sentences), and “the current issue covered by this review (one sentence)” should all be included in the background before moving on to “the objectives”. I would like the author to provide background information, a problem statement, and “their reasoning for branching off in this subsection”. The brief review section concludes with a phrase that places this subsection in a broader context. The conclusion should begin with one sentence that summarizes the main message using words like "Here we highlight." The authors should describe “the potential and the advancement this review” has made in the field in the first sentence of the conclusion, followed by “two to three sentences that provide a broader perspective” that is easily understood by a scientist from any discipline [2–5]. What is “MD”?
2. Keywords: Please double check that the keywords are listed in MeSH: Please list ten keywords chosen from Medical Subject Headings (MeSH) and use as many as possible in the title and in the first two sentences of the abstract [6].
3. Introduction: Please expand this section accordingly: I would like the authors to reorganize this section with “about 1000 words” and several paragraphs, introduce information on the key study constructs that should be understood by readers in any discipline, and “make it persuasive enough to advance the primary goal of the author's recent research and the particular goal the author has intended by this review.” I would like to suggest that the authors present the introduction beginning with the overall context, moving on to the specific context, and concluding with “the current problem addressed in this review” before moving on to “the objectives.” Those key structures ought to be set up logically and coherently [7] I also recommend that “the authors provide the rationale for presenting subsequent sections” in order to assist the reader in navigating the document.
4. Discussion: As suggested previously, I expect the authors to fully develop their arguments in this section by presenting crucial elements: I would like the authors to present the independent discussion section by opening with an introductory paragraph and followed by a summary of the previous sections. Then, I expect the authors to develop arguments clarifying the potential of this study as an extension of the previous work, the implication of the findings of this study, how this study could facilitate future research, the ultimate goal, the challenge, the knowledge and technology necessary to achieve this goal, the statement about this field in general, and finally the importance of this line of research. It is particularly important to present the limits, merit, and potential translation of this study to clinical practice [8,9].
5. Conclusion: This section is expected to be more than a summary. Please follow the guidelines I suggested previously: I think that presenting the conclusion would benefit from a single paragraph presenting some thoughtful as well as in-depth considerations by the authors as experts to convey the take-home message. The authors should make an effort to explain the theoretical implications as well as the translational application of their research. I believe that it would be necessary to discuss theoretical and methodological avenues in need of refinement as well as suggestions for a path forward in understanding the importance of this study.
6. References: Please follow the guidelines of the journal [1]. Please correct the font type.
Overall, the manuscript contains two figures, one table, and 201 references. I believe that the manuscript may have important value in presenting a valuable resource for anyone interested in the immunopathogenesis of MS. It provides a detailed overview of the latest research on the role of plasmablasts and pathogenic T cells in the development of the disease and highlights the current challenges and future directions in the field. I hope that, after these careful revisions, the manuscript can meet the journal’s high standards for publication. I am available for a new round of revisions to this article. I hope that, after these careful revisions, this paper can meet the journal’s high standards for publication. I am available for a new round of revisions to this article.
I declare no conflict of interest regarding this manuscript.
Best regards,
Reviewer
References:
- https://www.mdpi.com/journal/biologics/instructions
- https://www.scribbr.com/dissertation/abstract/
- https://writing.wisc.edu/handbook/assignments/writing-an-abstract-for-your-research-paper/
- https://doi.org/10.5812/ijem.100159
- https://doi.org/10.4103/sja.SJA_685_18
- https://meshb.nlm.nih.gov/
- https://dept.writing.wisc.edu/wac/writing-an-introduction-for-a-scientific-paper/
- https://doi.org/10.3163/1536-5050.103.2.001
- https://www.scribbr.com/dissertation/discussion/
16 August 2023
Manuscript ID: biologics-2487904
Type: Review
Title: “Immunopathogenesis of Multiple Sclerosis with Plasmablasts and Pathogenic T Cells” by Matsuzaka Y & Yashiro R, submitted to Biologics
Dear Authors,
After evaluating the document, it is clear that some minor revisions are necessary for the English language. The document contains grammatical errors. Moreover, certain sentences are unclear and require rephrasing to enhance readability. Although the content of the document is informative and well-organized, the quality of the English language needs improvement to ensure clarity and conciseness. Therefore, it is essential to perform minor editing to enhance the overall quality and readability of the document.
Best regards,
Reviewer
Author Response
Type: Review
Title: “Immunopathogenesis of Multiple Sclerosis with Plasmablasts and Pathogenic T Cells” by Matsuzaka Y & Yashiro R, submitted to Biologics
Dear Authors,
I am glad to see that the authors have made an effort to revise the manuscript. Regarding my review report, however, only partial revisions have been made. Prior to publication, I kindly request that the authors carefully consider my comments from the previous report. Here, I highlight unrevised sections with quotation marks "..." to assist the authors in revising the manuscript to meet the journal's high standards. Also, I anticipate the authors preparing a detailed rebuttal to my remarks.
Comments:
- Abstract: Please make an effort to present main components proportionally: I suggest the authors present “the background”, “a short summary”, and “a conclusion” in proportional order within the 200 words, according to the journal’s guidelines [4].The general background (one to two sentences), the specific background (two to three sentences), and “the current issue covered by this review (one sentence)” should all be included in the background before moving on to “the objectives”. I would like the author to provide background information, a problem statement, and “their reasoning for branching off in this subsection”. The brief review section concludes with a phrase that places this subsection in a broader context. The conclusion should begin with one sentence that summarizes the main message using words like "Here we highlight." The authors should describe “the potential and the advancement this review” has made in the field in the first sentence of the conclusion, followed by “two to three sentences that provide a broader perspective” that is easily understood by a scientist from any discipline [2–5]. What is “MD”?
- Keywords: Please double check that the keywords are listed in MeSH: Please list ten keywords chosen from Medical Subject Headings (MeSH) and use as many as possible in the title and in the first two sentences of the abstract [6].
- Introduction: Please expand this section accordingly: I would like the authors to reorganize this section with “about 1000 words” and several paragraphs, introduce information on the key study constructs that should be understood by readers in any discipline, and “make it persuasive enough to advance the primary goal of the author's recent research and the particular goal the author has intended by this review.” I would like to suggest that the authors present the introduction beginning with the overall context, moving on to the specific context, and concluding with “the current problem addressed in this review” before moving on to “the objectives.” Those key structures ought to be set up logically and coherently [7] I also recommend that “the authors provide the rationale for presenting subsequent sections” in order to assist the reader in navigating the document.
- Discussion: As suggested previously, I expect the authors to fully develop their arguments in this section by presenting crucial elements: I would like the authors to present the independent discussion section by opening with an introductory paragraph and followed by a summary of the previous sections. Then, I expect the authors to develop arguments clarifying the potential of this study as an extension of the previous work, the implication of the findings of this study, how this study could facilitate future research, the ultimate goal, the challenge, the knowledge and technology necessary to achieve this goal, the statement about this field in general, and finally the importance of this line of research. It is particularly important to present the limits, merit, and potential translation of this study to clinical practice [8,9].
- Conclusion: This section is expected to be more than a summary. Please follow the guidelines I suggested previously: I think that presenting the conclusion would benefit from a single paragraph presenting some thoughtful as well as in-depth considerations by the authors as experts to convey the take-home message. The authors should make an effort to explain the theoretical implications as well as the translational application of their research. I believe that it would be necessary to discuss theoretical and methodological avenues in need of refinement as well as suggestions for a path forward in understanding the importance of this study.
- References: Please follow the guidelines of the journal [1]. Please correct the font type.
Overall, the manuscript contains two figures, one table, and 201 references. I believe that the manuscript may have important value in presenting a valuable resource for anyone interested in the immunopathogenesis of MS. It provides a detailed overview of the latest research on the role of plasmablasts and pathogenic T cells in the development of the disease and highlights the current challenges and future directions in the field. I hope that, after these careful revisions, the manuscript can meet the journal’s high standards for publication. I am available for a new round of revisions to this article. I hope that, after these careful revisions, this paper can meet the journal’s high standards for publication. I am available for a new round of revisions to this article.
I declare no conflict of interest regarding this manuscript.
Best regards,
According to reviewer’s comment, we added some sentences in abstract section on line 11 to 15, 18 to 20, and 27 to 31, in page 1 to clear the background, a short summary, a conclusion, the objectivesand the current issue, in which main issue is as follows; “So far, steroid and immunosuppressants have been the mainstream for autoimmune diseases, but the problem is that it kills not only pathogenic T cells, but also lymphocytes that are necessary for the body.” If it were insufficient as abstract section, please let me know.